# Multimeric structure enables the acceleration of KaiB-KaiC complex formation induced by ADP/ATP exchange inhibition

Shin-ichi Koda[1,2]*, Shinji Saito[1,2]*

**1** Department of Theoretical and Computational Molecular Science, Institute for Molecular Science, Okazaki, Aichi, Japan, **2** School of Physical Sciences, The Graduate University for Advanced Studies (SOKENDAI), Okazaki, Aichi, Japan

\* koda@ims.ac.jp (SK); shinji@ims.ac.jp (SS)

**Data Availability Statement:** The source code and data used to produce the results and analyses presented in this manuscript are available from GitHub repository: https://github.com/shin1koda/kaibc_binding.

## Abstract

Circadian clocks tick a rhythm with a nearly 24-hour period in a variety of organisms. In the clock proteins of cyanobacteria, KaiA, KaiB, and KaiC, known as a minimum circadian clock, the slow KaiB-KaiC complex formation is essential in determining the clock period. This complex formation, occurring when the C1 domain of KaiC hexamer binds ADP molecules produced by the ATPase activity of C1, is considered to be promoted by accumulating ADP molecules in C1 through inhibiting the ADP/ATP exchange (ADP release) rather than activating the ATP hydrolysis (ADP production). Significantly, this ADP/ATP exchange inhibition accelerates the complex formation together with its promotion, implying a potential role in the period robustness under environmental perturbations. However, the molecular mechanism of this simultaneous promotion and acceleration remains elusive because inhibition of a backward process generally slows down the whole process. In this article, to investigate the mechanism, we build several reaction models of the complex formation with the pre-binding process concerning the ATPase activity. In these models, six KaiB monomers cooperatively and rapidly bind to C1 when C1 binds ADP molecules more than a given threshold while stabilizing the binding-competent conformation of C1. Through comparison among the models proposed here, we then extract three requirements for the simultaneous promotion and acceleration: the stabilization of the binding-competent C1 by KaiB binding, slow ADP/ATP exchange in the binding-competent C1, and relatively fast ADP/ATP exchange occurring in the binding-incompetent C1 in the presence of KaiB. The last two requirements oblige KaiC to form a multimer. Moreover, as a natural consequence, the present models can also explain why the binding of KaiB to C1 reduces the ATPase activity of C1.

## Author summary

Circadian clocks tick a rhythm with a nearly 24-hour period in various organisms. The cyanobacterial circadian clock, composed of only three proteins, KaiA, KaiB, and KaiC, has attracted much attention because of its simplicity. The slow KaiB-KaiC complex

**Funding:** This work was funded by JSPS KAKENHI, Grant Number JP18K14185 to SK and JP21H04676 to SS, https://www.jsps.go.jp/. The funders had no role in study design, data collection and analysis, decision to publish, or preparation of the manuscript.

**Competing interests:** The authors have declared that no competing interests exist.

formation is essential in determining the clock period in this system. Significantly, this complex formation is accelerated together with its promotion, implying a potential role in the period-keeping mechanism. However, this simultaneous promotion and acceleration is theoretically exceptional because this complex formation is promoted by inhibiting a backward process, which generally slows down the whole process. In this article, we investigate the molecular mechanism of this phenomenon by building mathematical models to find that the binding of KaiB to C1 must lower the ADP/ATP exchange of the C1 domain of KaiC via stabilizing the binding-competent conformation of C1. The present results would be of benefit for future investigations on functional roles of the simultaneous promotion and acceleration in the KaiABC oscillator.

## Introduction

Circadian clocks are endogenous timing systems. Organisms ranging from bacteria to higher plants and animals use the clocks to adapt their activity to daily changes in the environment. The cyanobacterial circadian clock has attracted much attention because it oscillates without transcription-translation feedback [1] and can be reconstituted *in vitro* by mixing three proteins, KaiA, KaiB, and KaiC, in the presence of ATP [2].

KaiC, the main component of the oscillator, forms a homohexamer consisting of N-terminal C1 and C-terminal C2 rings [3, 4]. In the presence of KaiA and KaiB, two amino acid residues near the ATP binding site in C2, Ser431 and Thr432, are periodically phosphorylated and dephosphorylated [5, 6]. During this oscillation, KaiA facilitates the phosphorylation by acting on C2 [7–10]. After C2 is phosphorylated, KaiB, on the other hand, inhibits the KaiA activity on C2 [11, 12] by forming the stable C1-KaiB-KaiA complex [13–15]. Then, C2 returns to the unphosphorylated state.

Keeping period constant is a fundamental function of biological clocks. For understanding the period-determining mechanism, slow processes of the KaiABC oscillator have intensively been studied. For example, the binding of KaiB to C1, taking a few hours, affects the period of the KaiABC oscillator as well as that of the damped oscillation in the absence of KaiA *in vivo* [16]. This KaiB-C1 complex formation proceeds in a dual conformational selection manner, i.e., both KaiB and C1 need to change their conformations into binding-competent ones before the binding [14, 15]. Thus, there have been attempts to determine the slower process of the two. An experimental study has reported a slow KaiB-KaiC complex formation [17]. This study has explained the slowness by assuming a slow conformational transition of KaiB. In contrast to this experiment conducted at only one protein concentration, another experiment has measured the KaiB concentration dependence of the binding rate [18]. Remarkably, in light of the kinetic theory [19], the result indicates that the pre-binding conformational transition of KaiC is the rate limiting process. Moreover, we have theoretically shown that the slow KaiB-KaiC binding of the former experiment can be explained by the attractive KaiB-KaiB interaction in the ring-shaped KaiB-KaiC complex, without assuming the slow conformational transition of KaiB [20]. Thus, it is now likely that the slowness of the binding arises from KaiC rather than KaiB. Moreover, the KaiB-KaiC complex formation plays another significant role in clock output and is still being actively studied [21–24].

To understand the whole picture of the KaiB-KaiC complex formation, it is then necessary to reveal the detail of the pre-binding processes of KaiC. Several experiments have shown that the ATPase activity of C1 closely relates to the complex formation. For example, a mutation

disabling the ATP hydrolysis of C1 [25] or introduction of a non-hydrolyzable ATP analogue inhibits the complex formation [18, 25]. Moreover, the binding-competent conformation of KaiC binds ADP in C1 [14, 15, 26]. These results imply that the conformational transition to the binding-competent conformation occurs after the ATP hydrolysis of C1. This picture of the KaiC's pre-binding process can be briefly summarized as

$$\text{C1}_{\text{ATP}} \rightleftharpoons \text{C1}_{\text{ADP}} \rightleftharpoons \text{C1}_{\text{ADP}}^*, \tag{1}$$

where the subscripts represent the nucleotide binding to C1, and the asterisk indicates the binding-competent conformation. This type of pre-binding process involving the ATPase activity can be seen in a molecularly detailed reaction model [27], for example. Note that, as pointed out by Paijmans *et al.* [27] (see the next section for detailed review), the population of the binding-competent conformation is likely controlled by the exchange rate of bound ADP for exogenous ATP rather than the rate of ATP hydrolysis (the backward and forward processes of $\text{C1}_{\text{ATP}} \rightleftharpoons \text{C1}_{\text{ADP}}$ in Eq (1), respectively). To put it another way, the phosphorylation of C2 shifts the equilibrium of Eq (1) toward the forward direction by inhibiting the backward ADP/ATP exchange of C1.

In the present article, we focus on the binding rate of KaiB to C1, which determines the time required for the complex formation. A recent experiment reported that a phosphomimetic mutant of KaiC with a high KaiB affinity, S431E/T432E-KaiC (KaiC-EE), shows a faster KaiB-KaiC complex formation than the wild-type KaiC (KaiC-WT) [16]. That is, the more the complex formation is promoted due to the phosphorylation of C2, the more accelerated (i.e., the apparent binding rate constant increases). We call this phenomenon the simultaneous promotion and acceleration of the complex formation. The simultaneous promotion and acceleration may have a significant role in the period robustness against environmental perturbation such as temperature because it can prevent excess phosphorylation of C2 from taking time by accelerating the complex formation, which terminates the phosphorylation. Thus, the molecular mechanism of this simultaneous promotion and acceleration of the KaiB-KaiC complex formation is worth investigating in detail. However, the simple scheme in Eq (1) cannot explain it. Under this scheme, the ADP/ATP exchange inhibition reduces the apparent binding rate because the relaxation rate of $\text{C1}_{\text{ATP}} \rightleftharpoons \text{C1}_{\text{ADP}}$ is given by the sum of the rates of the ATP hydrolysis and ADP/ATP exchange. This contradiction with the experimental result implies that the scheme in Eq (1) is oversimplified or lacks some important factors.

In the following section, to clarify the molecular mechanism of the simultaneous promotion and acceleration, we build several reaction models of the KaiB-KaiC complex formation with pre-binding processes of KaiC, which can be seen as an extension of the scheme in Eq (1). The present models explicitly consider the hexameric form and are derived from assumptions based on experimental results. Specifically, reflecting that $\text{KaiB}_6\text{KaiC}_6$ complex is highly more stable than $\text{KaiB}_n\text{KaiC}_6$ ($n < 6$) due to the attractive adjacent KaiB-KaiB interaction [15, 20, 28, 29], the present models assume that six KaiB monomers immediately bind to C1 only when C1 binds ADP molecules more than a given threshold. Then, through comparison among the models proposed here, we extract several requirements for the simultaneous promotion and acceleration and discuss structure-function relations of it. Additionally, as a corollary from the present models, we further show that the binding of KaiB to C1 reduces the ATPase activity of C1 [30], which has not been explained by any model so far.

## Results

### Model setup

We here build several reaction models of the binding of KaiB to the C1 ring of a KaiC hexamer so that we can explain the simultaneous promotion and acceleration of the KaiB-KaiC complex formation [16].

**Hexameric model with independent C1 monomers.** We firstly build a model that explicitly considers the hexameric form of KaiC and the six nucleotides binding to C1. We denote KaiC that binds $n$ ATP and $(6 - n)$ ADP in C1 by $C_{6,n}$ ($0 \leq n \leq 6$). Because the ATP binding sites of KaiC are almost always occupied by ATP or ADP [31], we ignore KaiC that binds less than six nucleotides in C1. We consider the ATP hydrolysis as a transition from $C_{6,n}$ to $C_{6,n-1}$ and the ADP/ATP exchange as the backward process. We ignore the ATP synthesis due to its high activation energy.

In this hexameric model, we impose the following five assumptions on the KaiB-KaiC complex formation.

- (i) The C1 domain of each monomer in a KaiC hexamer has binding-competent and binding-incompetent conformations, and the conformational transition between them undergoes independently of the other monomers.

- (ii) The conformational transition to the binding-competent conformation is feasible only in a monomer binding ADP.

- (iii) The binding-competent conformation is negligibly populated in the absence of KaiB but is stabilized by the binding of KaiB.

- (iv) The complexes between a KaiC hexamer and less than six KaiB monomers are negligible, while $KaiB_6KaiC_6$ complex is highly stable.

- (v) The conformational transitions of C1 and KaiB and the association/dissociation of KaiB to/from KaiC are much faster than the ATP hydrolysis and the ADP/ATP exchange of C1.

These five assumptions are supported by the following reasons. Assumption (i) is consistent with the experimental result that several crystal structures of the C1 ring consist of multiple conformations of the monomer [26], suggesting that each monomer is not strongly constrained by the other monomers. Assumption (ii), which is also adopted by the scheme in Eq (1), is based on the experimental result that the KaiB-KaiC complex binds ADP in C1 [14, 15, 26]. Assumption (iii) is consistent with the experimental result that the addition of KaiB increases the binding-competent conformation of C1 [18]. Although a small amount of the binding-competent conformation may exist in the real system even in the absence of KaiB, we ignore it for simplicity. Note that this assumption is a typical case of the conformational-selection binding mechanism, where the binding stabilizes a less populated energetically excited conformation. Assumption (iv) is based on the experimental observations on the stoichiometry of the KaiB-KaiC complex [28, 29], where the adjacent KaiB-KaiB interaction stabilizes the complex [15, 20]. The stability of $KaiB_6KaiC_6$ complex is also suggested by the experimental result that the phosphorylation oscillation is unaltered by an increase in KaiB concentration [32], indicating that the binding between KaiB and binding-competent KaiC is already saturated around the standard concentrations of Kai proteins. Lastly, Assumption (v) is consistent with the experimental result that the ATPase activity of C1 affects the period [26, 30], which implies that processes involved in the ATPase activity are the rate limiting process in the complex formation. Although the processes assumed fast here could have comparable rates to the ATPase activity, we focus only on the ATPase activity for simplicity in this model.

With these assumptions, we construct the present model as follows. In the absence of KaiB, due to Assumptions (ii) and (iii), KaiC takes only the binding-incompetent conformation in the model. Thus, in this case, the present model consists of the ATP hydrolysis and the ADP/ATP exchange of C1 among binding-incompetent KaiC, $C_{6,n}$ ($0 \leq n \leq 6$). More explicitly, the model is represented as

$$C_{6,6} \underset{k_e}{\overset{6k_h}{\rightleftharpoons}} C_{6,5} \underset{2k_e}{\overset{5k_h}{\rightleftharpoons}} C_{6,4} \underset{3k_e}{\overset{4k_h}{\rightleftharpoons}} C_{6,3} \underset{4k_e}{\overset{3k_h}{\rightleftharpoons}} C_{6,2} \underset{5k_e}{\overset{2k_h}{\rightleftharpoons}} C_{6,1} \underset{6k_e}{\overset{k_h}{\rightleftharpoons}} C_{6,0}, \tag{2}$$

where $k_h$ and $k_e$ are the rate constants of the ATP hydrolysis and the ADP/ATP exchange of a binding-incompetent C1 monomer, respectively.

In the presence of KaiB, KaiB binds to C1 cooperatively, strongly, rapidly, and only when C1 binds six ADP

$$C_{6,0} + 6B \rightarrow C_{6,0}^* B_6^*, \tag{3}$$

where B and B* represents a binding-incompetent and binding-competent KaiB monomers, respectively, and $C_{6,0}^*$ is a KaiC hexamer consisting of six binding-competent monomers. The strong cooperative binding of six KaiB, where its backward process can be disregarded, arises from Assumption (iv). This cooperative binding occurs only when C1 binds six ADP because all the six KaiC monomers in a KaiC hexamer need to bind ADP to take the binding-competent conformation due to Assumptions (i) and (ii). Since Assumption (v) requires the conformational transitions of KaiB and KaiC and the KaiB-KaiC binding to be fast, the process in Eq (3) immediately proceeds once $C_{6,0}$ is produced in the presence of abundant KaiB. On the other hand, $C_{6,0}^* B_6^*$ immediately dissociates once a bound ADP is exchanged into ATP because KaiC with less than six ADP in C1 cannot form a complex in the present model.

To sum up, the whole binding scheme in the presence of abundant KaiB is given by

$$C_{6,6} \underset{k_e}{\overset{6k_h}{\rightleftharpoons}} C_{6,5} \underset{2k_e}{\overset{5k_h}{\rightleftharpoons}} C_{6,4} \underset{3k_e}{\overset{4k_h}{\rightleftharpoons}} C_{6,3} \underset{4k_e}{\overset{3k_h}{\rightleftharpoons}} C_{6,2} \underset{5k_e}{\overset{2k_h}{\rightleftharpoons}} C_{6,1} \underset{6k_e^*}{\overset{k_h}{\rightleftharpoons}} C_{6,0}^* B_6^*, \tag{4}$$

where $k_e^*$ is the rate constant of the ADP/ATP exchange of a binding-competent C1 monomer. Eq (4) differs from Eq (2) in the last transition between $C_{6,1}$ and $C_{6,0}^* B_6^*$, which represents the immediate formation and dissociation of the KaiB-KaiC complex mentioned above. Their rates are dominated by the ATP hydrolysis and the ADP/ATP exchange of C1, respectively.

**Hexameric models with cooperative C1 monomers.**   It must be noted that the crystal structures indicating the independent conformational transition of each monomer (Assumption (i)) [26] are C2-truncated C1 hexamers. Hence, the properties of the C1 ring of the full-length KaiC may differ from the C2-truncated C1 ring. Moreover, as C2 has certain cooperativity [33, 34], it is no wonder that C1, homologous to C2, also acts cooperatively. Therefore, we consider a cooperative conformational transition of C1 with modified assumptions:

- (i') The C1 domain of each monomer in a KaiC hexamer has binding-competent and binding-incompetent conformations, and the conformational transition between them undergoes cooperatively together with all the six monomers.

- (ii') The conformational transition to the binding-competent conformation is feasible when C1 binds $n$ ($1 \leq n \leq 6$) or more ADP.

With these modified assumptions and Assumptions (iii-v), the binding scheme in the presence of abundant KaiB is modified as follows (only the scheme with $n = 4$ is shown, for

example):

$$\mathrm{C}_{6,6} \underset{k_\mathrm{e}}{\overset{6k_\mathrm{h}}{\rightleftharpoons}} \mathrm{C}_{6,5} \underset{2k_\mathrm{e}}{\overset{5k_\mathrm{h}}{\rightleftharpoons}} \mathrm{C}_{6,4} \underset{3k_\mathrm{e}}{\overset{4k_\mathrm{h}}{\rightleftharpoons}} \mathrm{C}_{6,3} \underset{4k_\mathrm{e}^*}{\overset{3k_\mathrm{h}}{\rightleftharpoons}} \mathrm{C}_{6,2}^* \mathrm{B}_6^* \underset{5k_\mathrm{e}^*}{\overset{2k_\mathrm{h}^*}{\rightleftharpoons}} \mathrm{C}_{6,1}^* \mathrm{B}_6^* \underset{6k_\mathrm{e}^*}{\overset{k_\mathrm{h}^*}{\rightleftharpoons}} \mathrm{C}_{6,0}^* \mathrm{B}_6^*, \tag{5}$$

where $k_\mathrm{h}^*$ is the rate constant of the ATP hydrolysis of a binding-competent C1 monomer. In a similar manner to the original scheme in Eq (4), six KaiB monomers immediately bind to C1 and stabilize the binding-competent conformation of C1 when C1 binds $n$ or more ADP. We hereafter refer to this scheme as "Model $n$." Note that Model 6 is apparently identical to the original one in Eq (4). Thus, we refer to the original one as Model 6 henceforth, although they are derived from different assumptions.

Before moving on to the next topic, we here clarify relations of the present hexameric models to the framework of the Monod–Wyman–Changeux (MWC) allosteric model [35], which describes multimeric effects on conformational transitions as seen in several previous hexameric models of the KaiABC oscillator [27, 34]. In the present study, the conformational transition feasibility depending on bound nucleotide is expressed by the six discrete models with a definite threshold number of bound ADP required for the transition. By contrast, the MWC framework can describe the bound nucleotide dependence in one unified continuous model by introducing a parameter defining the strength of the dependence. In the MWC framework, the equilibrium constant between the binding-incompetent and binding-competent conformation $K_\mathrm{conf}$ can be given, for example, by

$$K_\mathrm{conf} = K_\mathrm{conf,0}\exp[-\mu n_\mathrm{ADP}], \tag{6}$$

where $n_\mathrm{ADP}$ is the number of bound ADP in C1, $K_\mathrm{conf,0}$ is the equilibrium constant when $n_\mathrm{ADP} = 0$, and $\mu$ is the strength parameter for bound nucleotide dependence. Importantly, the present six hexameric models can be regarded as special cases of the MWC model with large $\mu$ and proper $K_\mathrm{conf,0}$ despite the difference in appearance (see Methods for details). In the present study, we dare to avoid the unified continuous description of the MWC-based model and instead individually describe the six discrete models because the discrete description is suitable for qualitative understanding of the mechanism of the phenomenon of interest, as shown below.

**Monomeric model.**   To clarify the roles of the hexameric form of KaiC, we further consider for comparison a simplified monomeric model consisting of the following schemes:

$$\mathrm{C}_\mathrm{ATP} \underset{k_\mathrm{e}}{\overset{k_\mathrm{h}}{\rightleftharpoons}} \mathrm{C}_\mathrm{ADP}, \tag{7}$$

$$\mathrm{C}_\mathrm{ATP} \underset{k_\mathrm{e}^*}{\overset{k_\mathrm{h}}{\rightleftharpoons}} \mathrm{C}_\mathrm{ADP}^* \mathrm{B}^*, \tag{8}$$

where $\mathrm{C}_\mathrm{ATP}$ and $\mathrm{C}_\mathrm{ADP}$ are KaiC monomers that bind ATP and ADP in C1, respectively.

**Overview of the KaiB-KaiC complex formation.**   The experimental observation shows that, after adding KaiB to KaiC in its steady state, a part of KaiB immediately binds to KaiC and then more KaiB-KaiC complex is gradually formed [16]. For describing this behavior in the present models, it is expected that

$$k_\mathrm{e} > k_\mathrm{e}^* \ \text{or} \ k_\mathrm{h} < k_\mathrm{h}^*. \tag{9}$$

If this relation is satisfied, the KaiB-KaiC complex formation proceeds along with the following scenario. In Model $n$ ($1 \leq n \leq 6$), when abundant KaiB is added to a KaiC only system equilibrated according to Eq (2), KaiB immediately binds to initially existing

$C_{6,k}$ ($6 - n \leq k \leq 6$). Then, $C_{6,k}^* B_6^*$ gradually increases because the relation in Eq (9) shifts the equilibrium toward $C_{6,k}^* B_6^*$ in Eq (5). This initially rapid and subsequently slow binding is consistent with the experimental observation [16]. Note that the apparent binding rate is given by the relaxation rate of the scheme in Eq (5).

**C2 dependence of the rate constants of C1.** As mentioned in Introduction, phosphorylation of C2 promotes the KaiB-C1 complex formation. In this study, as seen in the mathematical model of Paijmans *et al.* [27], we adopt a framework of C1-C2 interaction where phosphorylation of C2 modulates the rate constants of the ATP hydrolysis and/or ADP/ATP exchange of C1 so that the population of the binding-competent C1 increases. A significant problem here is which rate constant of C1 and how it is modulated to accumulate bound ADP molecules in C1 for the conformational transition, in other words, whether the phosphorylation of C2 promotes the ATP hydrolysis or inhibits the ADP/ATP exchange of C1. Focusing on the effect on the ATPase activity, which increases in the former and decreases in the latter case, Paijmans *et al.* employ the latter in their model [27] because it is consistent with experimental results that a KaiC mutant mimicking the phosphorylated C2 shows a lower ATPase activity than the unphosphorylated KaiC-WT [30]. Following their study, we assume in the present study that only $k_e$ and $k_e^*$ are modulated according to the phosphorylation state of C2 while $k_h$ and $k_h^*$ are unaltered. Note that $k_h$ and $k_h^*$ could be somewhat affected by C2 in the real system. However, we ignore the phosphorylation dependence as a simplification in mathematical modeling.

## Acceleration of the binding by inhibiting the ADP/ATP exchange

This subsection numerically shows that Model $n$ ($2 \leq n \leq 6$) can describe the simultaneous promotion and acceleration of the complex formation [16] with proper parameters while the monomeric model and Model 1 fail. We further theoretically investigate the reason for the failure in the monomeric model and Model 1 to find that the multi-step ATPase activity originating from the multimeric structure of KaiC is essential for the simultaneous promotion and acceleration. On the other hand, since all of Model $n$ ($2 \leq n \leq 6$) can reproduce the experimental data of the complex formation well, the present result cannot determine the most suitable model for the real system. For this problem, we propose an experiment where observables depend highly on $n$.

**Comparison between the hexameric model (Model 6) and the monomeric model.** To clarify qualitative distinction between the monomeric and hexameric models, we firstly compare the monomeric model with an extreme case of the six hexameric models, Model 6.

To simulate the behaviors of KaiC-WT and KaiC-EE by the present model, we consider a set of rate constants, $k_h$, $k_{eWT}$, $k_{eWT}^*$, $k_{eEE}$, and $k_{eEE}^*$, where $k_h$ is the ATP hydrolysis rate constant common to KaiC-WT and KaiC-EE, and $k_{eWT(EE)}$, $k_{eWT(EE)}^*$ are the ADP/ATP exchange rate constants of the binding-incompetent and binding-competent KaiC-WT (KaiC-EE), respectively. As mentioned in the previous subsection, we assume that only $k_e$ and $k_e^*$ depend on the phosphorylation state of C2. Hence, $k_h$ is common to KaiC-WT and KaiC-EE here.

First, we examine whether the present model can reproduce the experimental result of the KaiB-KaiC complex formation (Fig 1). We calculate the amount of KaiB binding to KaiC-WT with ($k_h$, $k_{eWT}$, $k_{eWT}^*$) and to KaiC-EE with ($k_h$, $k_{eEE}$, $k_{eEE}^*$) along with the hexameric scheme in Eq (4) and the monomeric scheme in Eq (8). The initial states are set to be the steady states of Eqs (2) and (7), respectively.

Since the hexameric model (Model 6) has a wide range of rate constants that satisfactorily reproduce the experimental data (Fig 1A), we sample their distributions by Bayesian parameter estimation with Markov chain Monte Carlo (MCMC), which is used in a variety of areas in

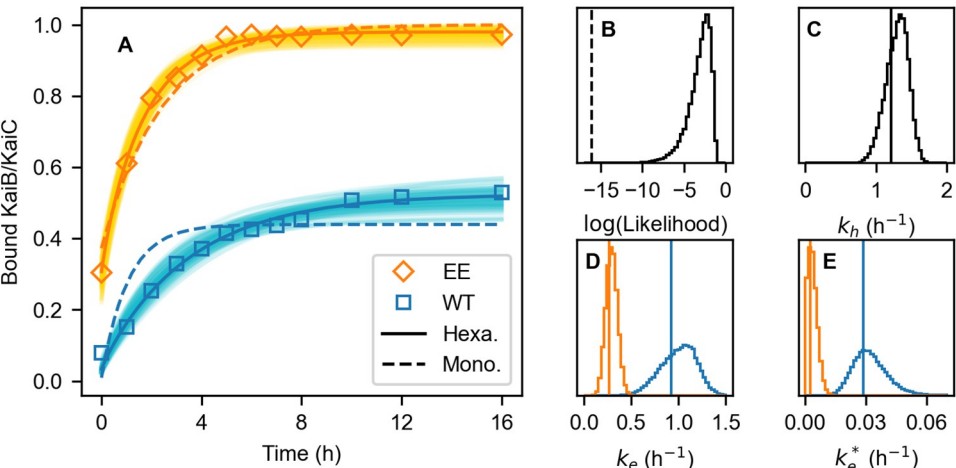

**Fig 1. Results of the Bayesian parameter estimation on the hexameric model (Model 6) and the monomeric model.** (A) Time courses of the binding of KaiB to KaiC-WT (blue) and KaiC-EE (orange). Squares are the experimental data. Bold solid and dashed curves are the best fits of the hexameric and monomeric models, respectively. Thin solid curves show 100 results of the hexameric model randomly chosen from the MCMC sampling. (B) Distribution of the hexameric model's likelihood logarithm, which is defined in Methods. The vertical dashed line shows the likelihood logarithm of the monomeric model's best fit. (C-E) Distributions of the hexameric model's rate constants: (C) ATP hydrolysis $k_h$, (D) ADP/ATP exchange of the binding-incompetent conformation $k_e$ of KaiC-WT (blue) and KaiC-EE (orange), and (E) ADP/ATP exchange of the binding-competent conformation $k_e^*$ of KaiC-WT (blue) and KaiC-EE (orange). Vertical lines show the corresponding best-fit values.

natural science [36], including kinetic models based on ordinary differential equations in systems biology [37–39] and, in particular, that regarding the KaiABC oscillator [40]. In the present sampling, to narrow the range of the parameters, we additionally use the experimental data of the ATPase activities of KaiC in the presence and absence of KaiB [30] (see Methods). This result shows that, as expected in Eq (9), both $k_{eWT}^*$ and $k_{eEE}^*$ are indeed smaller than $k_{eWT}$ and $k_{eEE}$, respectively (Fig 1D and 1E). Owing to these magnitude relations, the KaiB-KaiC complex formation proceeds as explained above. Moreover, both $k_{eEE}$ and $k_{eEE}^*$ are lowered compared to $k_{eWT}$ and $k_{eWT}^*$, respectively (Fig 1D and 1E), indicating that the hexameric model can simultaneously achieve the promotion and acceleration of the KaiB-KaiC complex formation by the C2's phosphorylation lowering $k_e$ and $k_e^*$.

On the other hand, the monomeric model fails to reproduce the KaiB bindings to KaiC-WT and KaiC-EE simultaneously (Fig 1A and 1B). This best fit only reproduces the promotion of the binding by inhibiting the ADP/ATP exchange and cannot reproduce the acceleration of the binding. Thus, the monomeric model cannot accelerate the binding by inhibiting the ADP/ATP exchange.

Next, we analyze the binding rate in detail. As mentioned in the previous subsection, the apparent binding rate is given by the relaxation rate, that is, the smallest non-zero eigenvalue of the transition matrix of the pre-binding process shown in Eqs (4) or (8). We denote them by $\gamma_{hexa}$ and $\gamma_{mono}$, respectively.

In the monomeric scheme, $\gamma_{mono}$ is given by the sum of the rate constants of the forward and backward processes in Eq (8) as

$$\gamma_{mono} = k_h + k_e^*. \qquad (10)$$

Thus, $\gamma_{mono}$ never increases by lowering $k_e$ or $k_e^*$. This is why the monomeric model cannot accelerate the binding by inhibiting the ADP/ATP exchange.

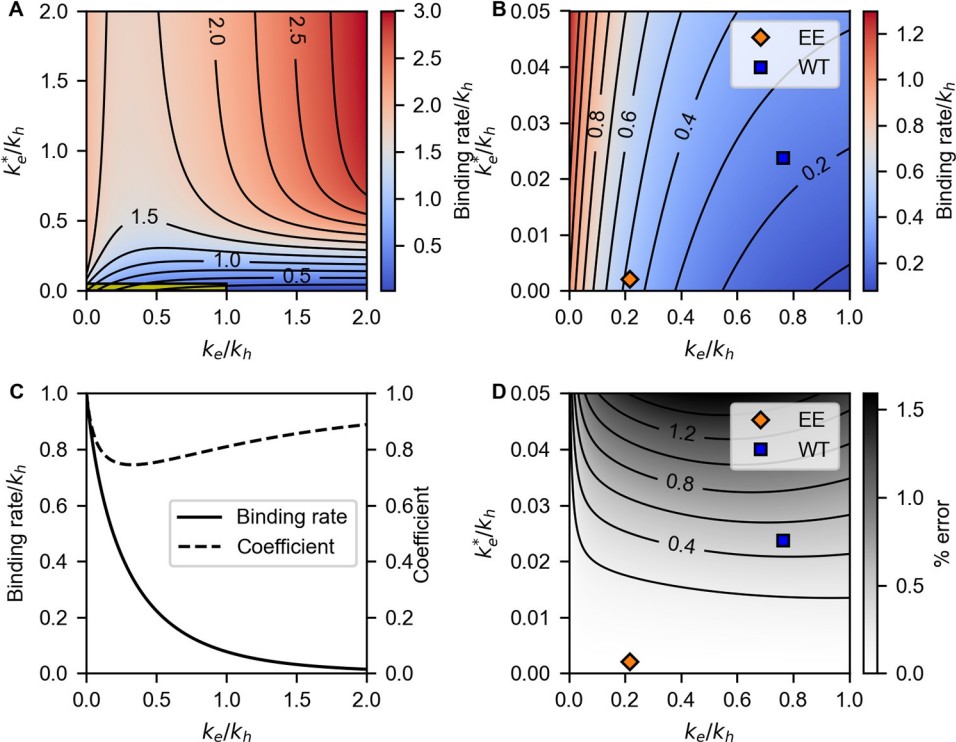

**Fig 2. Binding rate of the hexameric model (Model 6), $\gamma_{\text{hexa}}$.** All rate constants are normalized by $k_{\text{h}}$. (A-B) 2D plots of $\gamma_{\text{hexa}}$ as a function of $k_{\text{e}}$ and $k_{\text{e}}^*$. The yellow region in (A) is magnified in (B). Blue and orange squares correspond to the best-fit parameters of KaiC-WT and KaiC-EE. Contours are plotted every 0.25 in (A) and 0.1 in (B). (C) The relaxation rate of Eq (11) $\gamma_{\text{hexa}}$ (solid curve) and the coefficient $\alpha$ in Eq (12). (D) Percent error of the perturbative approximation of $\gamma_{\text{hexa}}$ shown in Eq (13). Blue and orange squares are the same as (B). Contours are plotted every 0.2%.

In the hexameric scheme, any analytical expression of $\gamma_{\text{hexa}}$ is not available. Instead, we numerically calculate $\gamma_{\text{hexa}}$ as a function of $k_{\text{e}}$ and $k_{\text{e}}^*$ (Fig 2A and 2B). This result indicates that $\gamma_{\text{hexa}}$ is an increasing function with respect to $k_{\text{e}}^*$ as well as $\gamma_{\text{mono}}$. However, in contrast to the monomeric scheme, $\gamma_{\text{hexa}}$ increases with decreasing $k_{\text{e}}$ when $k_{\text{e}}^*$ is small. With this property, KaiC-EE of the hexameric model achieves the simultaneous promotion and acceleration of the complex formation by lowering $k_{\text{e}}$ and $k_{\text{e}}^*$ (Fig 2B).

The next problem is why $\gamma_{\text{hexa}}$ increases with decreasing $k_{\text{e}}$ when $k_{\text{e}}^*$ is small. Since $k_{\text{e}}^*$ is small, we can approximate $\gamma_{\text{hexa}}$ by a perturbation with respect to $k_{\text{e}}^*$ (see Methods). The zeroth-order term of $\gamma_{\text{hexa}}$, which we denote by $\gamma_{\text{hexa},0}$, is given by the relaxation rate of

$$\text{C}_{6,6} \underset{k_{\text{e}}}{\overset{6k_{\text{h}}}{\rightleftharpoons}} \text{C}_{6,5} \underset{2k_{\text{e}}}{\overset{5k_{\text{h}}}{\rightleftharpoons}} \text{C}_{6,4} \underset{3k_{\text{e}}}{\overset{4k_{\text{h}}}{\rightleftharpoons}} \text{C}_{6,3} \underset{4k_{\text{e}}}{\overset{3k_{\text{h}}}{\rightleftharpoons}} \text{C}_{6,2} \underset{5k_{\text{e}}}{\overset{2k_{\text{h}}}{\rightleftharpoons}} \text{C}_{6,1} \overset{k_{\text{h}}}{\rightarrow} \text{C}_{6,0}^* \text{B}_6^*. \tag{11}$$

Because the final destination of this scheme is $\text{C}_{6,0}^* \text{B}_6^*$, $\gamma_{\text{hexa},0}$ can be regarded as a generalized rate constant from $\text{C}_{6,1-6}$ to $\text{C}_{6,0}^* \text{B}_6^*$. On the other hand, the first-order term of $\gamma_{\text{hexa}}$ is given by the relaxation rate of

$$\text{C}_{6,1} \overset{6\alpha k_{\text{e}}^*}{\longleftarrow} \text{C}_{6,0}^* \text{B}_6^*, \tag{12}$$

where $\alpha$ is the coefficient determined by the perturbation. Thus, $\gamma_{\text{hexa}}$ is approximately given

by the sum of the generalized rate constants of the forward and backward processes as

$$\gamma_{\text{hexa}} \simeq \gamma_{\text{hexa},0} + 6\alpha k_e^*. \tag{13}$$

Fig 2D shows that this expression well approximates $\gamma_{\text{hexa}}$ with less than 1.5% error around the optimized parameter set. We plot $\gamma_{\text{hexa},0}$ and $\alpha$ as a function of $k_e$ in Fig 2C. Because $\alpha$ does not change drastically with $k_e$, and because $k_e^*$ is small, the $k_e$ dependence of $\gamma_{\text{hexa}}$ is mainly determined by $\gamma_{\text{hexa},0}$. Moreover, as shown in Fig 2C, $\gamma_{\text{hexa},0}$ is a decreasing function of $k_e$ because the increase of $k_e$ inhibits the transition to $C_{6,0}^* B_6^*$ in Eq (11). Therefore, $\gamma_{\text{hexa}}$ increases with decreasing $k_e$ when $k_e^*$ is small.

**Comparison among the hexameric models.** Next, we investigate the reproducibility of Model 1-5 by Bayesian parameter estimation in a similar fashion to the analysis above. We regard the six rate constants in the schemes in Eqs (2) and (5), $k_h$, $k_h^*$, $k_{eWT}$, $k_{eWT}^*$, $k_{eEE}$, and $k_{eEE}^*$, as the parameters to be estimated. In the parameter estimation here, in addition to the constraints employed above to narrow the range of parameters, we further use the reduction of the ATPase activity after adding KaiB [30] as a reference experimental data (see Methods).

The results show that, while Model 2-5 successfully reproduce the experimental data on the complex formation [16] (Fig 3B–3E) as well as Model 6, Model 1 cannot describe the simultaneous promotion and acceleration of the complex formation (Fig 3A) as well as the

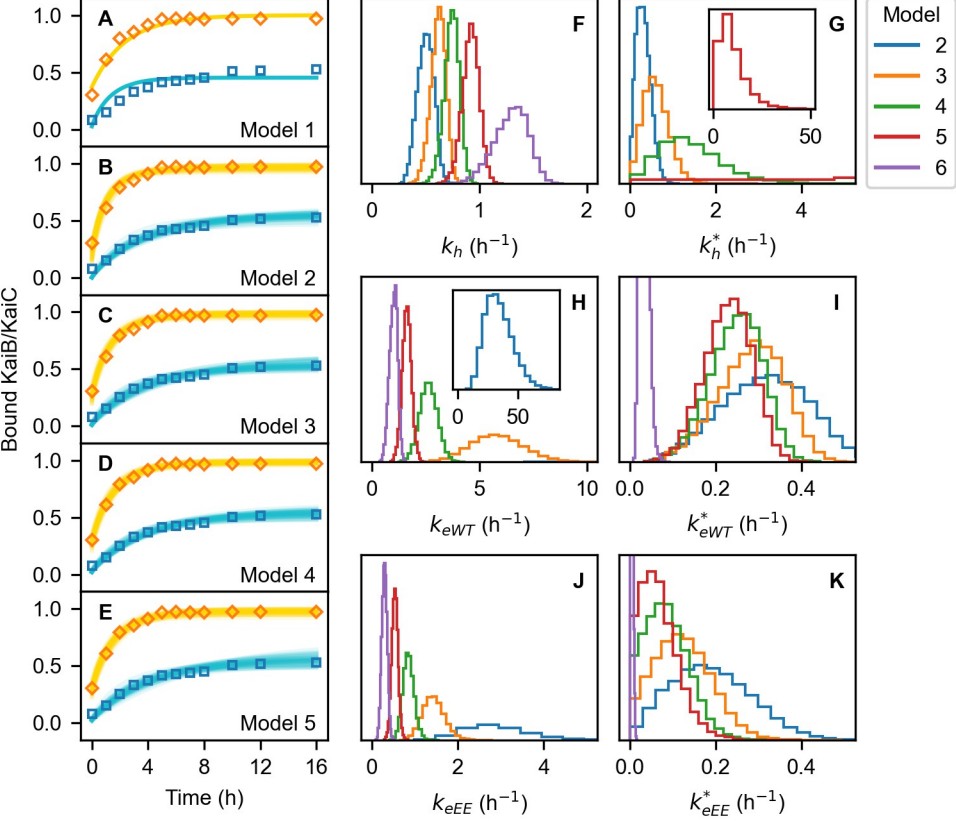

**Fig 3. Results of the Bayesian parameter estimation on Models 1-5.** (A-E) Time courses of the binding of KaiB to KaiC-WT (blue) and KaiC-EE (orange) of Models 1-5, respectively. Squares are the experimental data. Solid curves in (A) are the best fits of Model 1. In (B-E), 100 results of each model randomly chosen from the MCMC sampling are plotted in thin solid curves. (F-K) Distributions of the rate constants in Models 1-6: (F) $k_h$, (G) $k_{h^*}$, (H) $k_{eWT}$, (I) $k_{eWT}^*$, (J) $k_{eEE}$, and (K) $k_{eEE}^*$. (G) does not include Model 6 because $k_{h^*}$ is absent in this model.

monomeric model. These results can be analogically explained by the above comparison between Model 6 and the monomeric model. As can be seen from the distributions of the sampled rate constants (Fig 3F–3K), $k_{eWT}^*$ and $k_{eEE}^*$ are smaller than $k_{eWT}$ and $k_{eEE}$, respectively, as well as Model 6. Thus, the schemes of Models 2-5 can also be divide into the fast transitions from the binding-incompetent to -competent C1 and the slow reverse transitions, which are represented, for example in Model 4, as

$$C_{6,6} \underset{k_e}{\overset{6k_h}{\rightleftharpoons}} C_{6,5} \underset{2k_e}{\overset{5k_h}{\rightleftharpoons}} C_{6,4} \underset{3k_e}{\overset{4k_h}{\rightleftharpoons}} C_{6,3} \overset{3k_h}{\longrightarrow} C_{6,2}^* B_6^* \overset{2k_{h^*}}{\longrightarrow} C_{6,1}^* B_6^* \overset{k_{h^*}}{\longrightarrow} C_{6,0}^* B_6^* \tag{14}$$

and

$$C_{6,3} \overset{4k_{e^*}}{\leftarrow} C_{6,2}^* B_6^* \overset{5k_{e^*}}{\leftarrow} C_{6,1}^* B_6^* \overset{6k_{e^*}}{\leftarrow} C_{6,0}^* B_6^*, \tag{15}$$

respectively. Here again, decrease in $k_e$ accelerates the effective forward transitions in Eq (14) and eventually results in the simultaneous promotion and acceleration of the complex formation as in Model 6. On the other hand, the separation of fast and slow processes in Model 1 is given by

$$C_{6,6} \overset{6k_h}{\rightarrow} C_{6,5}^* B_6^* \overset{5k_{h^*}}{\rightarrow} \cdots \overset{k_{h^*}}{\rightarrow} C_{6,0}^* B_6^* \tag{16}$$

and

$$C_{6,6} \overset{k_e^*}{\leftarrow} C_{6,5}^* B_6^* \overset{2k_e^*}{\leftarrow} \cdots \overset{6k_e^*}{\leftarrow} C_{6,0}^* B_6^*. \tag{17}$$

In this case, the ADP/ATP exchange is absent in the forward transitions in Eq (16) as in the monomeric model. Therefore, inhibition of the ADP/ATP exchange only reduces the rate of the reverse processes in Eq (17), resulting in the deceleration of the complex formation.

The comparison between Models 1-6 and the monomeric model can be summarized into the following three key requirements for the simultaneous promotion and acceleration (Fig 4). First, the binding of KaiB to KaiC stabilizes the binding-competent conformation of C1, which is less populated in the absence of KaiB (highlighted in green in Fig 4). This requirement ensures that the addition of KaiB brings changes into the system due to property differences between the binding-incompetent and binding-competent C1. The second requirement is that

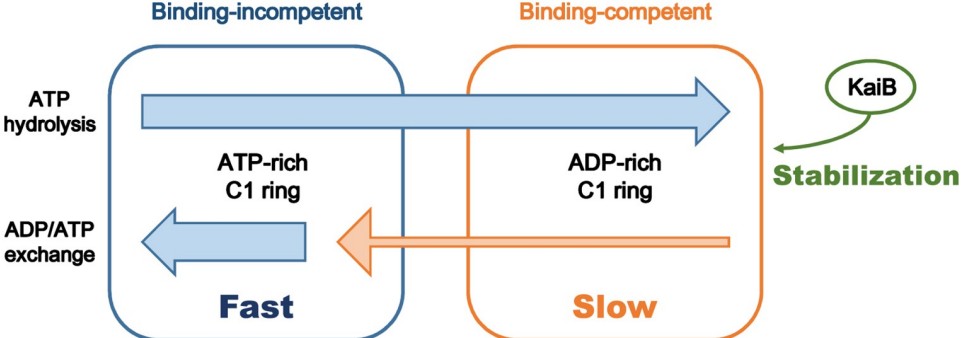

**Fig 4. Graphical summary of the keys for the simultaneous promotion and acceleration of the complex formation.** (green) Stabilization of the binding-competent C1 by KaiB. (orange) Slow ADP/ATP exchange in the binding-competent C1. (blue) Fast ADP/ATP exchange in the binding-incompetent C1 in the presence of KaiB.

the ADP/ATP exchange rate of the binding-competent C1 is slower than the other processes (orange thin arrow in Fig 4). This requirement has double roles. On the one hand, it promotes the transition to the binding-competent C1 via the bound ADP accumulation in C1 in the presence of KaiB, resulting in the further KaiB-KaiC complex formation. On the other hand, it effectively separates the processes in Eq (5) into the principal fast processes from binding-incompetent to binding-competent C1 (Eq (14)) and the minor slow backward process (Eq (15)). Finally, the third requirement is that relatively fast ADP/ATP exchange occurs in the binding-incompetent C1 even in the presence of KaiB (blue thick arrow in Fig 4). Due to this requirement, the effective forward processes toward the binding-competent C1 includes this ADP/ATP exchange as in Eq (14). Since this ADP/ATP exchange is a negative regulator of the effective forward processes, its inhibition by C2 phosphorylation results in the acceleration of the forward processes and consequently the whole process of the KaiB-KaiC complex formation.

Note that the second and third requirements need the ADP/ATP exchange both in the binding-incompetent and binding-competent C1 in the presence of KaiB, as in Eqs (14) and (15). To satisfy these requirements, KaiC must form a multimer to have a multi-step ATPase activity. In other words, the multimeric structure of KaiC enables the simultaneous promotion and acceleration of the KaiB-KaiC complex formation.

**On the complex formation rates in the model by Paijmans *et al.*** Since we have clarified the requirements for the simultaneous promotion and acceleration of complex formation, we then see if other previous mathematical models of the KaiABC oscillator can achieve it. We here investigate the model by Paijmans *et al.* [27] because it explicitly describes the interplay between the ATPase activity and the KaiB-KaiC complex formation. Although Paijmans' model is for the whole KaiABC oscillator, it can also simulate the binding of KaiB to phospho-mimetic KaiC mutants by fixing the phosphorylation state of C2, i.e., setting the rate constants of the (de)phosphorylation reactions to zero. Since Paijmans' model is based on the MWC allosteric model [35], in which the present hexameric models are also rooted, it is expected that proper choice of the parameters in Paijmans' model enables the simultaneous promotion and acceleration of complex formation.

First, we examine the default parameter set [27]. In Paijmans' model, the ADP/ATP exchange rate of C1 is determined through Eqs (11) and (12) of the original paper [27]. Since the default parameter set obliges the ADP/ATP exchange rate constant of the inactive C1 (corresponding to the present binding-competent conformation) to be always larger than or equal to that of the active (binding-incompetent) C1, the default set does not satisfy the second requirement for the simultaneous promotion and acceleration. To confirm that the default parameter set is incompatible with the simultaneous promotion and acceleration, we simulate the complex formation between KaiB and KaiC in which $m$ ($0 \leq m \leq 6$) monomers are in D (doubly phosphorylated) state and ($6 - m$) monomers are in U (unphosphorylated) state (Fig 5A) and then calculate the apparent binding rate by fitting the curves in Fig 5A to a single exponential function (Fig 5C). As expected, the results clearly show that the binding rate decreases as the binding is promoted along with the phosphorylation. Next, we conduct the same calculations with parameters modified so that the ADP/ATP exchange rate of the inactive (binding-competent) C1 is smaller than that of the active (binding-incompetent) C1 (see Methods for details). The requirements for the simultaneous promotion and acceleration are satisfied in this case, and the results indeed exhibit the expected behavior (Fig 5B and 5C). The comparison between the default and modified parameter sets further support the requirements for the simultaneous promotion and acceleration of complex formation.

Note that the modified parameter set used above concentrates only on achieving the simultaneous promotion and acceleration. Unfortunately, it lacks many other significant properties

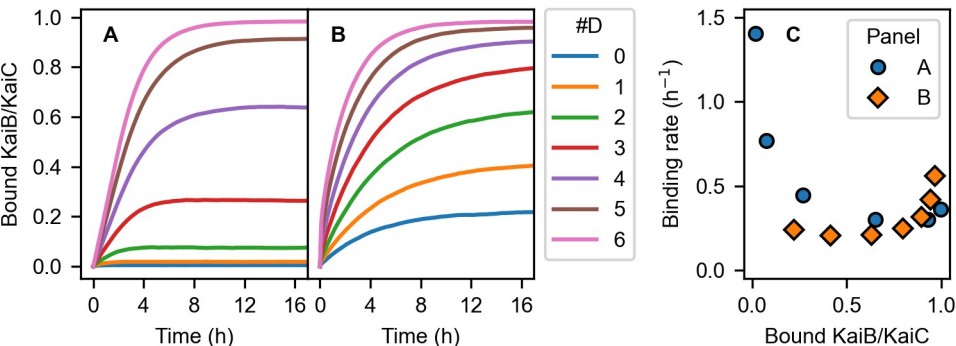

**Fig 5. KaiB-KaiC complex formation in Paijmans' model.** (A,B) Time courses of the binding of KaiB to KaiC in which $m$ ($0 \leq m \leq 6$) monomers are in D (doubly phosphorylated) state and ($6 - m$) monomers are in U (unphosphorylated) state calculated with the default parameters in the original paper [27] (A) and with the modified parameters (B). (C) Correlations between the amount of the complex formation and the apparent binding rate. Blue circles and orange squares are calculated from the curves in (A) and (B), respectively. The data on the fully dephosphorylated KaiC in (A) is omitted because of its faint amount of the complex formation.

of the KaiABC oscillator, such as the oscillation itself. Thus, it is necessary to search for a rhythmic parameter set that sufficiently describes the KaiB-KaiC complex formation in case further functional investigations are planned with Paijmans' model.

**Proposal to estimate the number of ADP required for the binding-competent C1.** The parameter estimation above shows that Models 2-6 can reproduce the experimental data on the complex formation [16] with high precision (Fig 3B–3E). This means, however, that further experimental investigation is needed to determine the most suitable model for describing the real system. One way is to measure the rate constants of the elementary processes in C1, such as $k_h$, $k_h^*$, $k_{eWT}$, $k_{eWT}^*$, $k_{eEE}$, and $k_{eEE}^*$. However, direct measurement of elementary processes is generally difficult when a number of processes are tangled. What is worse, many of the rate constants of the present models have certain overlap each other (Fig 3F–3K). For this problem of the direct measurement, we here propose an experiment where observables vary according to the models.

Let us recall the experiments on KaiC hexamer in which two mutants coexist, prepared through monomerization, mixing, and re-hexamerization [33, 34]. Then, consider this time the mixing of KaiC-EE and that with additional mutations disabling the ATPase activity of C1 such as E77Q and E78Q (hereafter KaiC-C1cat⁻-EE). It is expected in this experiment that if C1 requires many bound-ADP for the conformational transition as in Model 6, even slight contamination of ATPase-disabled KaiC-C1cat⁻-EE decreases the binding-competent C1. On the other hand, if C1 requires just a few bound-ADP for the conformational transition as in Model 2, the binding-competent C1 allows the contamination to some extent. Fig 6 shows the final amount of bound KaiB to C1 of each model with respect to the contamination of KaiC-C1cat⁻-EE, calculated with the sampled rate constants (see Methods for detailed calculation). As expected, the binding feasibility is vulnerable against the contamination of KaiC-C1-cat⁻-EE when the threshold number of bound ADP for the complex formation is large (Model 6), while it is robust when the threshold is low (Model 2). This result shows a clear distinction depending on the threshold number and would be of benefit in determining the threshold.

## Reduction of the ATPase activity of C1 by the complex formation

As a corollary from the KaiB-KaiC complex formation explained above, we here explain why the complex formation reduces the ATPase activity (ADP production rate) of C1 in the present

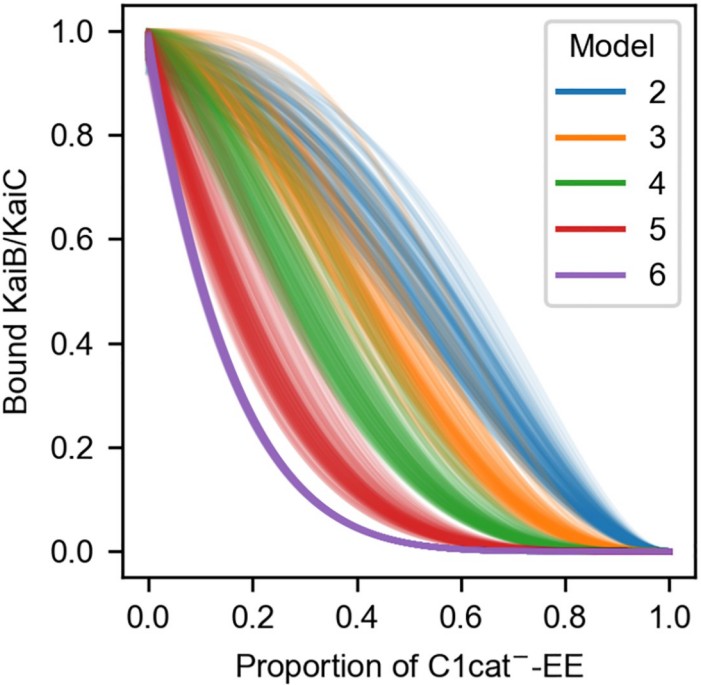

**Fig 6. Expected final amount of bound KaiB with respect to the contamination of KaiC-C1cat⁻-EE.** 100 sets of the rate constants are randomly chosen from the sampling in each model.

model. The essential mechanism can be clearly seen in Model 6. In Model 6, the ATPase activity of C1 par KaiC monomer $\gamma_{\mathrm{ATPase}}$ is represented as

$$\gamma_{\mathrm{ATPase}} = \frac{k_{\mathrm{h}}}{C_{\mathrm{tot}}} \sum_{n=1}^{6} n[C_{6,n}]. \tag{18}$$

Since $\sum_{n=1}^{6} n[C_{6,n}]$ is equal to the concentration of bound ATP in C1, the equilibrium shift in Eq (4) by the complex formation reduces $\gamma_{\mathrm{ATPase}}$.

Note that the reduction of the ATPase activity has not been explained by any mathematical model so far. The most significant factor in the present model is the difference between the ADP/ATP exchange rate of the binding-incompetent and binding-competent C1. In the present model, the binding of KaiB to C1 stabilizes the binding-competent C1 with a low ADP/ATP exchange rate and thus reduces bound ATP in C1. On the other hand, other previous mathematical models of the KaiABC oscillator do not consider such C1 or KaiB binding dependence of the ADP/ATP exchange rate of C1. For example, Paijmans' model [27] with the default parameters has an inverse magnitude relation of the ADP/ATP exchange, as mentioned above. This is why the KaiB binding does not reduce the ATPase activity in the previous models.

With Model 6 and parameters sampled above, we calculate the time courses of the ATPase activity of C1 in the presence of KaiB (Fig 7A). Since the initial state is the steady state of the KaiC only system, the initial ATPase activity corresponds to that in the absence of KaiB. Although the values of the ATPase activity are widely distributed (Fig 7B and 7C), the reduction by the complex formation is common in all cases (Fig 7A) as explained above.

In Models 1-5, the situation becomes slightly complex because of the ATP hydrolysis in the binding-competent C1. Since $k_{\mathrm{h}}^{*}$ is basically larger than $k_{\mathrm{h}}$ (Fig 3F and 3G), the binding of KaiB

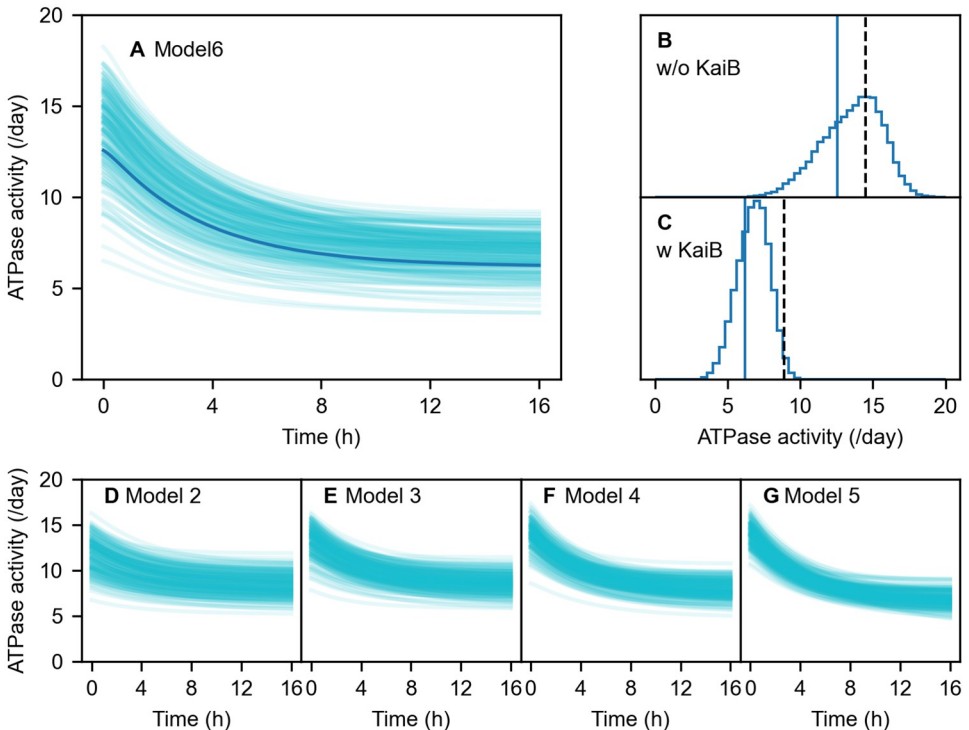

**Fig 7. Results of the Bayesian parameter estimation on the ATPase activity of C1.** (A) Time courses of the ATPase activity of C1 during the KaiB-KaiC complex formation in Model 6. Bold curve corresponds to the best-fit parameters. Thin solid curves show 200 results randomly chosen from the MCMC sampling. (B) and (C) Distributions of the ATPase activities in Model 6 in the absence (B) and presence (C) of KaiB. Solid vertical lines correspond to the best-fit parameters of the hexameric model. Dashed vertical lines show the experimental data of the ATPase activities of the whole KaiC. (D-G) Time courses of the ATPase activity of C1 during the KaiB-KaiC complex formation in Models 2-5. In each model, 200 results are randomly chosen from the MCMC sampling.

also results in promoting ATP hydrolysis via the stabilization of the binding-competent C1. However, on account of the ATPase activity reduction in total, this increasing effect is considered marginal, and hence, the decreasing effect due to the slower ADP/ATP exchange in the binding-competent C1 is dominant also in this case. To keep the increasing effect small, we impose an additional restriction on the ATPase activity in the parameter estimation for Models 2-5. With this restriction, the ATPase activity reduction is adequately described (Fig 7D–7G).

## Discussion

In the present study, we focused on the simultaneous promotion and acceleration of the KaiB-KaiC complex formation by the phosphorylation of C2, that is, both the amount of the KaiB-KaiC complex formation and its apparent binding rate increase simultaneously as C2 is phosphorylated [16]. As suggested by structural studies [14, 15, 26], this complex formation proceeds when C1 binds ADP molecules. Hence, the ATP hydrolysis and ADP/ATP exchange in C1 serve as essential pre-binding processes of C1 because they can control the bound-nucleotide state of C1. In particular, the complex formation is considered to be promoted by inhibiting the ADP/ATP exchange rather than by activating the ATP hydrolysis [27]. In this case, the acceleration of the complex formation, on the other hand, requires particular mechanism to achieve it because inhibition of a backward process generally decelerates the whole process.

To explain the molecular mechanism of the simultaneous promotion and acceleration, we proposed several hexameric models (Models 1-6) that explicitly consider the six bound nucleotides in C1 (Eqs (2), (4) and (5)). We assumed in Model $n$ that six KaiB monomers cooperatively and rapidly binds to C1 with the stabilization of the binding-competent conformation of C1 only when C1 binds $n$ or more ADP molecules (Assumptions (i', ii', and iii-v)). Among these assumptions, the cooperativity in the KaiB-KaiC complex formation arises from Assumptions (ii') and (iv); the former (ii') hypothetically defines a cooperative conformational transition of C1, and the latter (iv) emphasizes the cooperative binding of six KaiB based on the attractive KaiB-KaiB interaction [15, 20]. Although all the present six hexameric models arise from Assumptions (i', ii', and iii-v), it should be noted that Model 6 can be alternatively derived from Assumptions (i-v). Importantly, the cooperativity originates only from KaiB in this case because Assumption (ii) allows the C1 domain of each KaiC monomer in a hexamer to change its conformation independently of the other monomers, as indicated by the crystal structure of the C1 ring [26]. Therefore, Assumption (iv) is a logically minimum requirement for the cooperativity in the KaiB-KaiC complex formation.

Through the parameter estimations with Models 1-6 and the monomeric model, we found the three key requirements for the simultaneous promotion and acceleration of the complex formation (Fig 4): stabilization of the binding-competent C1 by the binding of KaiB, slow ADP/ATP exchange in the binding-competent C1, and relatively fast ADP/ATP exchange in the binding-incompetent C1 in the presence of KaiB. The most significant point is that the second requirement effectively separates the pre-binding ATPase activity into fast and slow parts. As a result, the fast ATP hydrolysis and ADP/ATP exchange in the binding-incompetent C1 compose an effective pre-binding process toward the binding-competent conformation, where the ADP/ATP exchange serves as a negative regulator. The inhibition of the ADP/ATP exchange by C2 phosphorylation then accelerates the forward processes and, consequently, the whole process of the KaiB-KaiC complex formation.

If some of the three requirements are not satisfied, it is impossible to achieve the simultaneous promotion and acceleration of the complex formation. For example, the present Model 1 and monomeric model violate the third requirement. Due to this violation, the (effective) forward process toward the binding-competent conformation consists only of the ATP hydrolysis, and hence, is never accelerated by inhibiting ADP/ATP exchange. Indeed, they fail to describe the simultaneous promotion and acceleration (Figs 1A and 3A). Another example is the default parameter set of Paijmans' model [27], which does not satisfy the second requirement. The complex formation, in this case, becomes slow along with C2 phosphorylation (Fig 5A and 5C). These results numerically validate the requirements for the simultaneous promotion and acceleration.

In realizing the simultaneous promotion and acceleration of the complex formation, the multimeric structure of KaiC plays an essential role. The second and third requirements need the ADP/ATP exchange both in the fast and slow parts of the pre-binding process in the presence of KaiB, as in Eqs (14) and (15). To satisfy these requirements, KaiC must form a multimer to have the multi-step ATPase activity.

It should also be noted that the present reaction model is the first to propose that the ADP/ATP exchange rate of C1 depends not only on the phosphorylation of C2 but on the conformational state of C1 as $k_e^* < k_e$. This relation explains the simultaneous promotion and acceleration of the KaiB-KaiC binding, as well as the reduction of the ATPase activity in the presence of KaiB [30]. The present results suggest that the conformational state of C1 might be considered separately from that of C2, which highly depends on the phosphorylation of C2 [9, 10, 13, 41]. More precisely, the structural change of C2 is induced by the phosphorylation regardless of the presence of KaiB. By contrast, the present results suggest that the conformational change

of C1 becomes only feasible by the phosphorylation of C2 and is induced by the binding of KaiB. A simple way to account for the different triggers for the structural changes of C1 and C2 is to distinguish these changes as distinct ones. Regarding mathematical models of the KaiABC oscillator, previous models, however, have considered only one global conformational change of a whole KaiC hexamer, where C1 and C2 are fully interlocked [27, 34, 42]. For deeper understanding of the molecular mechanism of the oscillator, it would be beneficial to consider further the interplay between conformational transitions of C1 and C2.

For future work, it would be helpful to summarize the relation between present models and other mathematical models. As mentioned already, the present six discrete models are essentially included in the framework of the MWC allosteric model [35], which can, by contrast, continuously express the bound-ADP dependence of the complex formation feasibility by introducing a parameter. The present study adopted the discrete description for qualitative understanding of the phenomenon of interest to find, for example, that the failure in the extreme case of Model 1 leads to the third requirement for the simultaneous promotion and acceleration, and that the simpler Model 6, where $k_h^*$ is absent, already grasps the essence of the complex formation. However, the unified continuous description of the MWC framework would be technically useful when implementing the simultaneous promotion and acceleration in models for the whole KaiABC oscillator. One possible way to implement is to develop Paijmans' model [27], which is the only full model of the oscillator explicitly considering ATPase activity as elementary processes. This model also rooted in the MWC model and shown to be compatible with the simultaneous promotion and acceleration as long as a proper parameter is chosen (Fig 5B and 5C). However, it is still elusive whether this model can describe the simultaneous promotion and acceleration with other properties of the oscillator, such as the phosphorylation oscillation. Thus, careful consideration should be needed.

Although the present model explains the simultaneous promotion and acceleration of the complex formation, its significance in the KaiABC oscillator remains elusive. However, for example, it can be expected that this property plays a role in the period robustness against environmental perturbation such as temperature compensation of period by preventing excessive phosphorylation of C2 from taking time by terminating it early through the acceleration of the KaiB-KaiC complex formation. Thus, further functional investigations on the simultaneous promotion and acceleration would be of importance. Moreover, besides detailed molecular mechanisms of the KaiABC oscillator, it is also important to clarify universal clock properties among species. A recent theoretical study shows that the feedback mechanism given by the coupling between phosphorylation of C2 and sequestration of KaiA by KaiB has a mathematical similarity to biological clocks of other species [43]. In this context, it may be of value to investigate in the other systems the corresponding process to the simultaneous promotion and acceleration of the KaiB binding, which mediates the phosphorylation and sequestration in the cyanobacterial clock.

## Methods

### Derivation of Models 1-6 based on the MWC framework

Models 1-6 of the present study can be formally derived from the framework of MWC as follows.

First, we consider the equilibrium of the conformational transitions of KaiB and KaiC and the binding of KaiB to C1. Since we assume in the present study that these processes are much faster than the ATPase activity of C1 (Assumption (v)), we treat these fast processes by the rapid equilibrium approximation, in which the fast processes are immediately equilibrated while the reset of slow processes is stopped. Specifically, the following processes are assumed

to be equilibrated:

$$B \rightleftharpoons B^*, \tag{19}$$

$$C_{6,6-n_D} \rightleftharpoons C^*_{6,6-n_D}, \tag{20}$$

$$C^*_{6,6-n_D} + 6B^* \rightleftharpoons C^*_{6,6-n_D}B^*_6, \tag{21}$$

where $n_D$ is the number of bound ADP in C1. We ignore the complex formations involving the binding-incompetent KaiB or KaiC by assuming their dissociation constants as infinity, without loosing the thermodynamical consistency. We denote the equilibrium or dissociation constants of these processes by $K_{confB} (= [B]/[B^*])$, $K_{confC,n_D} (= [C_{6,6-n_D}]/[C^*_{6,6-n_D}])$, and $K_d$ $k_d (= [B^*]^6 [C^*_{6,6-n_D}]/[C^*_{6,6-n_D}B^*_6])$, respectively. In the spirit of the MWC model, we represent the $n_D$ dependence of $K_{confC,n_D}$ as

$$K_{confC,n_D} = K_{confC,0} \exp[-\mu n_D]. \tag{22}$$

To satisfy Assumption (iii), we assume that

$$K_{confC,n_D} \gg 1. \tag{23}$$

Under this assumption, $[C^*_{6,6-n_D}]$ is negligibly small at an equilibrium. The proportion of $[C^*_{6,6-n_D}B^*_6]$ in the total concentration of $C_{6,6-n_D}$ (including both binding-incompetent and -competent ones) at an equilibrium, $\alpha^*(n_D)$, is represented as

$$
\begin{aligned}
\alpha^*(n_D) \quad &= \frac{[B^*_6]^6}{[B^*_6]^6 + K_d(1 + K_{confC,n_D})} \\
&\simeq \frac{1}{1 + \dfrac{K_d K_{confC,0}}{[B^*_6]^6} \exp[-\mu n_D]} \\
&= \frac{1}{1 + \exp[-\mu(n_D - \xi)]},
\end{aligned} \tag{24}
$$

where

$$\xi = \frac{1}{\mu} \log\left( \frac{K_d K_{confC,0}}{[B^*_6]^6} \right). \tag{25}$$

Note that, $\alpha^*(n_D)$ becomes a step function incremented around $n_D = \xi$ if

$$\mu \gg 1. \tag{26}$$

Model $n$ of the present study can be obtained by letting $(\mu, K_d, K_{confC,0})$ to satisfy Eq (23) for all $n_D$ $(0 \leq n_D \leq 6)$, Eq (26), and

$$n - 1 < \xi < n. \tag{27}$$

In this case, $\alpha^*(n_D)$ is nearly zero when $0 \leq n_D < n$ and is nearly unity when $n \leq n_D \leq 6$, meaning that this MWC-based system is reduced to Model $n$ shown in Eq (5).

## Bayesian parameter estimation

In the framework of Bayesian parameter estimation, the estimation result is represented in the form of the conditional (posterior) probability of the rate constants $\mathbf{k}$ given the observed experimental data $\mathbf{y}_{\mathrm{exp}}$, which we denote by $p(\mathbf{k}|\mathbf{y}_{\mathrm{exp}})$. The basic rule of conditional probability known as Bayes' rule yields an expression of $p(\mathbf{k}|\mathbf{y}_{\mathrm{exp}})$ by the conditional probability called the likelihood function $p(\mathbf{y}_{\mathrm{exp}}|\mathbf{k})$ and the prior probability of $\mathbf{k}$, $p(\mathbf{k})$:

$$p(\mathbf{k}|\mathbf{y}_{\mathrm{exp}}) = \frac{p(\mathbf{y}_{\mathrm{exp}}|\mathbf{k})p(\mathbf{k})}{p(\mathbf{y}_{\mathrm{exp}})}, \tag{28}$$

where $p(\mathbf{y}_{\mathrm{exp}}) = \int d\mathbf{k}\, p(\mathbf{y}_{\mathrm{exp}}|\mathbf{k})p(\mathbf{k})$.

The prior probability $p(\mathbf{k})$ is determined by a priori knowledge of $\mathbf{k}$. Since we only know that $k$ is positive and not too large, we assume that $p(k)$ for each $k \in \{k_{\mathrm{h}}, k_{\mathrm{h}^*}, k_{\mathrm{eWT}}, k_{\mathrm{eWT}}^*, k_{\mathrm{eEE}}, k_{\mathrm{eEE}}^*\}$ is the uniform distribution on the interval $[0, 100]$ $(\mathrm{h}^{-1})$.

We assume for the likelihood function $p(\mathbf{y}_{\mathrm{exp}}|\mathbf{k})$ that, when $\mathbf{k}$ is given, each experimental data $y_{\mathrm{exp},i}$ obeys the normal distribution with mean $y_{\mathrm{model},i}(\mathbf{k})$ and standard deviation $\sigma_i$, where $y_{\mathrm{model},i}(\mathbf{k})$ is the corresponding value obtained from the model with the given $\mathbf{k}$. Thus, the logarithm of $p(\mathbf{y}_{\mathrm{exp}}|\mathbf{k})$ is given by

$$\log p(\mathbf{y}_{\mathrm{exp}}|\mathbf{k}) = -\frac{1}{2}\sum_i [y_{\mathrm{exp},i} - y_{\mathrm{model},i}(\mathbf{k})]^2/\sigma_i^2 + \text{constant}, \tag{29}$$

where the index $i$ runs over all the time points and the type of KaiC (i.e., KaiC-WT and KaiC-EE) of the data on the KaiB-KaiC complex formation. Although $\sigma_i$ should also be estimated as well as $\mathbf{k}$ for rigorous parameter estimation, we fix all $\sigma_i$ to 0.05 in the present study since the aim is to get a qualitative behavior of the model.

In the parameter estimation of the hexameric models, we additionally consider the ATPase activities of KaiC in the presence and absence of KaiB. Since the experimentally observed values consist of the contributions from both C1 and C2, they can be used as upper limits of the ATPase activity of C1 in the present model. Specifically, we add the following term to Eq (29):

$$-\frac{1}{2}\sum_i \min\{y_{\mathrm{exp},i}^{\mathrm{A}} - y_{\mathrm{model},i}^{\mathrm{A}}(\mathbf{k}), 0\}^2/\sigma_i^{\mathrm{A}2}, \tag{30}$$

where $y_{\mathrm{exp},i}^{\mathrm{A}}$ is the experimental data of the ATPase activity of the whole KaiC and $y_{\mathrm{model},i}^{\mathrm{A}}(\mathbf{k})$ is the model output of the ATPase activity of C1. The index $i$ runs over the two cases in the presence and absence of KaiB. For simplicity, we fix $(y_{\mathrm{exp},i}^{\mathrm{A}}, \sigma_i^{\mathrm{A}})$ to (8.9, 0.9) (/Day) in the presence of KaiB and to (14.5, 2.0) (/Day) in the absence of KaiB.

In the parameter estimation of Models 2-5, we further consider the difference of the ATPase activities of KaiC between in the presence and absence of KaiB. We add the following term to Eq (29):

$$-\frac{1}{2}(\Delta y_{\mathrm{exp}}^{\mathrm{A}} - \Delta y_{\mathrm{model}}^{\mathrm{A}})^2/\sum_i \sigma_i^{\mathrm{A}2}, \tag{31}$$

where $\Delta y_{\mathrm{exp}}^{\mathrm{A}}$ and $\Delta y_{\mathrm{model}}^{\mathrm{A}}$ are the difference of the experimental data and the model output, respectively.

We numerically sample the posterior distribution $p(\mathbf{k}|\mathbf{y}_{\mathrm{exp}})$ defined in Eq (28) by a MCMC method implemented in a Python package emcee (version 3.0.2) [44]. In this method, multiple walkers in $\mathbf{k}$-space are required. We first find the maximum point of $p(\mathbf{k}|\mathbf{y}_{\mathrm{exp}})$ by the optimization routine in SciPy (version 1.5.2). The rate equations are integrated by SciPy with the

LSODA method. Then, we prepare 32 walkers with their initial points created by adding random perturbation to the maximum point and sample 50000 points every chain. Following the procedure in the method implemented in emcee [45], we compute the autocorrelation times of the sampled chains $\tau_{\text{chain}}$. Then, we discard the initial 2 max$\{\tau_{\text{chain}}\}$ samples from each chain and thin the samples by choosing every 0.5min$\{\tau_{\text{chain}}\}$ sample.

## Perturbative approximation of the binding rate

We here briefly review the perturbative approximation of the eigenvalues of a matrix. For simplicity, we summarize only the case where a matrix $A$ is real and diagonalizable with real eigenvalues and eigenvectors. The transition matrix of first-order rate equations belongs to this class. We denote the matrix consisting of the right eigenvectors by $P$, its inverse matrix by $Q$, and the diagonalized matrix of $A$ by $D$, that is,

$$I = QP, \quad D = QAP. \tag{32}$$

We expand $A$, $P$, $Q$, and $D$ with respect to a small parameter $\epsilon$ like

$$A = \sum_{n=0} \epsilon^n A_n. \tag{33}$$

Comparison of the first-order terms of Eq (32) yields

$$Q_1 P_0 + Q_0 P_1 = O, \tag{34}$$

and

$$D_1 \quad = Q_0 A_1 P_0 + Q_1 A_0 P_0 + Q_0 A_0 P_1 \tag{35}$$

$$= Q_0 A_1 P_0 + Q_1 P_0 D_0 + D_0 Q_0 P_1 \tag{36}$$

$$= Q_0 A_1 P_0 + (Q_1 P_0) D_0 - D_0 (Q_1 P_0). \tag{37}$$

Because the diagonal elements of the second and third terms of the right hand side cancel out each other, the diagonal elements of $D_1$ coincide with those of $Q_0 A_1 P_0$. Thus, the first-order term of the perturbative approximation of an eigenvalue of $A$ is given by the "average" of $A_1$ by the corresponding right and left eigenvectors of $A_0$.

## Calculation with Paijmans' model

The parameters used in the present study are as follows. All of the rate constants of (de)phosphorylation reactions are set to $1.0 \cdot 10^{-6} \, \text{h}^{-1}$ to fix the phosphorylation state of C2. For the calculation for Fig 5B, we further modify several parameters as $\delta g_{\text{act,A}}^{\text{CI·ADP}}(\text{D}) = 0.2 \, \text{kT}$, $\delta g_{\text{act,I}}^{\text{CI·ADP}}(\text{U}) = 0.3 \, \text{kT}$, $\delta g_{\text{act,I}}^{\text{CI·ADP}}(\text{D}) = 0.5 \, \text{kT}$, and $k_{\text{off}}^{\text{CI·ADP}} = 2.5 \, \text{h}^{-1}$, and all the association and dissociation rates of KaiB are set to be 100 times larger than the default values.

The calculation is carried out as follows. We first prepare several initial states of KaiC in which $m$ ($0 \leq m \leq 6$) monomers are in D (doubly phosphorylated) state and $(6 - m)$ monomers are in U (unphosphorylated) state. Then, we run 20 hours of equilibration in the absence of KaiB (the association and dissociation rates of KaiB are set to be $1.0 \cdot 10^{-6}$ and $1.0 \cdot 10^2 \, \text{h}^{-1}$, respectively), followed by 20 hours of the production run in the presence of KaiB. All the quantities are averaged over 100 runs. The curve fitting in Fig 5C is done by the curve_fit method in SciPy using a single exponential function $f(x, a, b) = a(1 - e^{-bx})$.

## Complex formation of the mixture of KaiC-EE and KaiC-C1cat⁻-EE

Let the propotion of KaiC-C1cat⁻-EE monomer in total KaiC monomer be $p$ ($0 \leq p \leq 1$). We assume that the proportion of the hexamer that contains $m$ ($0 \leq m \leq 6$) KaiC-C1cat⁻-EE monomers, $C_6^m$, obeys the binomial distribution $B(6, p)$. Regarding $C_6^m$, Model $n$ ($2 \leq n \leq 6$) is modified as follows: the rate constants of the ATP hydrolysis of $C_6^m$ binding $k$ ($0 \leq k \leq 6$) ATP molecules, $C_{6,k}^m$, is given by $\max\{k-m, 0\}k_h$ or $\max\{k - m, 0\}k_h^*$, according to its conformational state because ATP hydrolysis is disabled in $m$ binding sites. Using this modified Model $n$, we calculate the final amount of bound KaiB to $C_6^m$ with a sampled parameter set and then take an average over $m$ with the weight $B(6, p)$. By so doing, we obtain the final amount of bound KaiB as a function of $(n, p)$, shown in Fig 6.

## Author Contributions

**Conceptualization:** Shin-ichi Koda, Shinji Saito.

**Data curation:** Shin-ichi Koda.

**Formal analysis:** Shin-ichi Koda.

**Funding acquisition:** Shin-ichi Koda, Shinji Saito.

**Investigation:** Shin-ichi Koda.

**Methodology:** Shin-ichi Koda.

**Project administration:** Shin-ichi Koda, Shinji Saito.

**Software:** Shin-ichi Koda.

**Supervision:** Shinji Saito.

**Validation:** Shin-ichi Koda.

**Visualization:** Shin-ichi Koda.

**Writing – original draft:** Shin-ichi Koda.

**Writing – review & editing:** Shinji Saito.

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
