## [Decision Letter · Decision Letter 0]

30 Aug 2021

Dear Dr. Koda,

Thank you very much for submitting your manuscript "Ring-shaped multimeric structure enables the acceleration of KaiB-KaiC complex formation induced by the ADP/ATP exchange inhibition" for consideration at PLOS Computational Biology.

As with all papers reviewed by the journal, your manuscript was reviewed by members of the editorial board and by several independent reviewers. In light of the reviews (below this email), we would like to invite the resubmission of a significantly-revised version that takes into account the reviewers' comments.

The reviewers acknowledge that the submitted modeling work investigates one of the fundamental mechanisms of KaiC regulating its binding with KaiB by focusing on the potential role of hydrolysis on the CI domain of KaiC. However, reviewers raise significant concerns about the assumptions of the proposed model, which are critical for the construction and interpretation of their model. Another reviewer strongly suggests integrating the proposed model into a comprehensive model and test consistency of their model with existing data and other published models.

We cannot make any decision about publication until we have seen the revised manuscript and your response to the reviewers' comments. Your revised manuscript is also likely to be sent to reviewers for further evaluation.

Sincerely,

Christian I. Hong, Ph.D.

Associate Editor

PLOS Computational Biology

Jason Haugh

Deputy Editor

PLOS Computational Biology

Your manuscript has been evaluated by three independent reviewers. The reviewers acknowledge that the submitted modeling work investigates one of the fundamental mechanisms of KaiC regulating its binding with KaiB by focusing on the potential role of hydrolysis on the CI domain of KaiC. However, reviewers raise significant concerns about the assumptions of the proposed model, which are critical for the construction and interpretation of their model. Another reviewer strongly suggests integrating the proposed model into a comprehensive model and test consistency of their model with existing data and other published models.

Reviewer's Responses to Questions

**Comments to the Authors:**

Reviewer #1: In this manuscript, the authors develop a computational model to suggest that KaiC promotes KaiB binding by reducing exchange of ADP for ATP rather than actively promoting ATP hydrolysis to generate the ADP-bound state. Fundamental to their model is the assumption that cooperative binding of KaiB to the CI domain of KaiC happens only when all 6 CI domains are bound to ADP. Their model relies on five assumptions about KaiB-KaiC binding that are rooted in large part from prior structural or biochemical studies. However, while most of these assumptions have a strong foundation in experimental data, one of them, (i), is based on a crystal structure of the isolated CI hexamer and therefore lacks potential regulation by the C-terminal CII domain, calling into question the reliability of their assumption that CI domains can take on KaiB binding-competent or -incompetent conformations independently of one another. Overall, this manuscript presents some interesting concepts about the fundamental properties of KaiC that regulate cooperative binding of KaiB that are worth publishing if the caveats or concerns below can be addressed.

Major:

1. Support for their first (i) assumption on lines 129-131, that C1 domains can independently assume binding-competent or -incompetent conformations in the hexamer, is based on a crystal structure of isolated CI hexamers from ref. 22 (i.e., in the absence of CII domains). There is potentially a major flaw in the model, as it neglects to consider a possible role for the CII domain (and its phosphostate) in regulating CI domain conformation through ring-ring stacking or some other mechanism. The possibility of a confounding factor in their model from the CII domain should be clearly stated.

2. The observation in prior work (ref. 16) that the phosphomimetic KaiC-EE binds to KaiC with an apparent increased binding rate compared to WT KaiC also implies that CII phosphorylation state plays an important role in KaiB binding, which has been shown many times before. It’s not clear how the authors come to the conclusion that the observation above suggests that inhibition of ADP/ATP exchange is the cause of acceleration in complex formation. Further support for this should be provided.

Minor:

1. The terminology used here can gets confusing (e.g., “inhibition of the backward process”). I would suggest using scientific terminology like ATP hydrolysis or product release to describe the forward and backward reactions instead.

2. On line 159, should it be “necessarily” instead of “necessary”?

Reviewer #2: The authors address the question of how KaiB binding is coupled to the ATPase cycle in the KaiC CI domain using mathematical modeling. This is in an important question because it is at the heart of the negative feedback loop that allows sustained oscillation in the cyanobacterial clock. The study reaches a number of interesting conclusions, including an intriguing alternative interpretation for the biphasic binding observed when KaiB is added to KaiC in several already published experiments.

However, I feel that the central conclusion, that cooperativity in KaiB binding to a KaiC hexamer allows for inhibition of the ATP exchange reaction to result in faster binding kinetics to be based on assumptions about the system that are too restrictive. While the proposed scenario, where a cooperativity mechanism can lead to acceleration of the relaxation time of a system in scenarios where a non-cooperative elementary reaction could only be slowed is theoretically quite interesting, I do not see a good reason for ignoring the possibility that the specific mechanism in KaiC involves regulation of both the ATP hydrolysis reaction and the exchange reaction. That is, a phosphorylation state favorable to binding could both accelerate hydrolysis in CI and inhibit subsequent exchange of ADP for ATP. In fact, existing data suggest that KaiC freshly loaded with ATP shows a burst of hydrolysis that might be consistent with such a scenario. The authors should revisit their analysis to consider this possibility. Is it a possible alternative explanation of the data?

Another point is that the scenario the authors proposed is quite extreme where stable binding of KaiB cannot occur unless it is at all 6 CI sites. i.e. a single catalytically dead subunit would completely prevent KaiB binding (is this plausible?). The authors should analyze whether the performance of the model degrades noticeably if the strict cooperativity is weakened.

A minor remark: the authors comment that because mutations that change the rate of the CI ATPase catalytic cycle affect the period, the other processes involved in KaiB-KaiC interaction are fast. This seems too strong. There could be other processes involved in period determination that are on a similar timescale to the CI catalytic cycle.

Reviewer #3: This manuscript by Koda and Saito presents a new mathematical model for part of the circadian clock in cyanobacteria. The cyanobacterial clock is fascinating since it can be reconstituted in vitro with KaiA, KaiB, and KaiC, as well as ATP. Many previous modeling papers have considered this system. However, this manuscript does seem to present a new part of the story in that it studies hydrolysis on the CI domain of KaiC. The CII domain has received much of the previous attention, yet we know that the CI domain and the binding of KaiB to sequester KaiA eventually is also an important step.

In general, the manuscript is well written. However, some work could be done in the results and methods sections to give placeholders for the readers who are not as familiar with the details of the cyanobacterial circadian clock.

Another benefit of the current manuscript is the use of Bayesian analysis. This is not new in cyanobacterial modeling (e.g., see the work of Rust, Dinner, and colleagues) but can indeed be a fundamental approach going forward.

My primary concern is that they focus on a small part of the whole picture. It is left to the reader to see precisely how this gets interpreted into 24-hour oscillations. So, I think the readers would want to see the results integrated into a fuller model for a journal like PLoS Computational Biology. All the system details may not be needed, but this would allow us to understand how the developed model is consistent with other models and the overall systems biology of the system.

Additionally, I think the authors should consider how phosphorylation in the CII domain affects their model of the CI domain further. Even if these events occur at different times in the cycle, some mechanism needs to be proposed for how information is spread across the whole protein. The authors hint at this in future work, but I think it may be required here to convince readers of the importance of the modeling.

**Have the authors made all data and (if applicable) computational code underlying the findings in their manuscript fully available?**

Reviewer #1: Yes

Reviewer #2: Yes

Reviewer #3: None

PLOS authors have the option to publish the peer review history of their article (what does this mean?). If published, this will include your full peer review and any attached files.

Reviewer #1: No

Reviewer #2: **Yes: **Michael Rust

Reviewer #3: No
---

## [Decision Letter · Decision Letter 1]

19 Feb 2022

Dear Dr. Koda,

Thank you very much for submitting your manuscript "Multimeric structure enables the acceleration of KaiB-KaiC complex formation induced by the ADP/ATP exchange inhibition" for consideration at PLOS Computational Biology. As with all papers reviewed by the journal, your manuscript was reviewed by members of the editorial board and by several independent reviewers. The reviewers appreciated the attention to an important topic. Based on the reviews, we are likely to accept this manuscript for publication, providing that you modify the manuscript according to the review recommendations.

Reviewers and I agree that the authors addressed most of the major concerns raised from the initial draft. A reviewer suggests to include more detailed discussion about different models of the Cyanobacterial clock , which will provide a better context of this manuscript.

Sincerely,

Christian I. Hong, Ph.D.

Associate Editor

PLOS Computational Biology

Jason Haugh

Deputy Editor

PLOS Computational Biology

[LINK]

Reviewers and I agree that the authors addressed most of the major concerns raised from the initial draft. A reviewer suggests to include more detailed discussion about different models of the Cyanobacterial clock , which will provide a better context of this manuscript.

Reviewer's Responses to Questions

**Comments to the Authors:**

Reviewer #1: In this revised manuscript, the authors develop a computational model to based on the idea that KaiC promotes KaiB binding by reducing exchange of ADP for ATP rather than actively promoting ATP hydrolysis to generate the ADP-bound state. Their updated models in this revised manuscript satisfactorily address concerns from the first round of review and better address uncertainties in the modeling. Changes to the text throughout the manuscript to address reviewer comments have also improved the manuscript. This body of work presents some interesting and testable hypotheses that should drive the field forward.

Minor:

1. The statement on line 56-58 should be supported with a citation of the KaiABC structures in refs. 14 and 15 (Snijder, J. et al. 2017, Science and Tseng, R. et al. 2017, Science).

2. The title might flow a bit better if ‘the’ was removed before “ADP/ATP exchange inhibition” or if it read “…induced by inhibition of ADP/ATP exchange.”

Reviewer #3: I think the authors have sufficient addresses the reviewer’s comments. It would have been good to have a more in depth discussion of the various models of the whole Cyanobacterial clock rather than just to reference Paijman’s model, especially as new simulations of the whole system were not added. Perhaps some of the other modeling could be discussed, for example the models now cited by the Rust group or the recently published model of Tyler et al. (Genome Biology)? At least a little more text about this could help.

**Have the authors made all data and (if applicable) computational code underlying the findings in their manuscript fully available?**

Reviewer #1: Yes

Reviewer #3: Yes

PLOS authors have the option to publish the peer review history of their article (what does this mean?). If published, this will include your full peer review and any attached files.

Reviewer #1: No

Reviewer #3: No

Figure Files:

Data Requirements:

Reproducibility:

References:

---

## [Editor Report · Decision Letter 2]

25 Feb 2022

Dear Dr. Koda,

We are pleased to inform you that your manuscript 'Multimeric structure enables the acceleration of KaiB-KaiC complex formation induced by ADP/ATP exchange inhibition' has been provisionally accepted for publication in PLOS Computational Biology.

Best regards,

Christian I. Hong, Ph.D.

Associate Editor

PLOS Computational Biology

Jason Haugh

Deputy Editor

PLOS Computational Biology

---

## [Editor Report · Acceptance letter]

2 Mar 2022

PCOMPBIOL-D-21-01171R2 

Multimeric structure enables the acceleration of KaiB-KaiC complex formation induced by ADP/ATP exchange inhibition

Dear Dr Koda,

I am pleased to inform you that your manuscript has been formally accepted for publication in PLOS Computational Biology. Your manuscript is now with our production department and you will be notified of the publication date in due course.

With kind regards,

Zsofia Freund
